# How Re-sampling Helps for Long-Tail Learning?

**Jiang-Xin Shi**[1]* **Tong Wei**[2]* **Yuke Xiang**[3] **Yu-Feng Li**[1]†

[1]National Key Laboratory for Novel Software Technology, Nanjing University, Nanjing, China
[2]School of Computer Science and Engineering, Southeast University, Nanjing, China
[3]Consumer BG, Huawei Technologies, Shenzhen, China
{shijx,liyf}@lamda.nju.edu.cn, weit@seu.edu.cn, yuke.xiang@huawei.com

## Abstract

Long-tail learning has received significant attention in recent years due to the challenge it poses with extremely imbalanced datasets. In these datasets, only a few classes (known as the *head classes*) have an adequate number of training samples, while the rest of the classes (known as the *tail classes*) are infrequent in the training data. Re-sampling is a classical and widely used approach for addressing class imbalance issues. Unfortunately, recent studies claim that re-sampling brings negligible performance improvements in modern long-tail learning tasks. This paper aims to investigate this phenomenon systematically. Our research shows that re-sampling can considerably improve generalization when the training images do not contain semantically irrelevant contexts. In other scenarios, however, it can learn unexpected spurious correlations between irrelevant contexts and target labels. We design experiments on two homogeneous datasets, one containing irrelevant context and the other not, to confirm our findings. To prevent the learning of spurious correlations, we propose a new *context shift augmentation* module that generates diverse training images for the tail class by maintaining a context bank extracted from the head-class images. Experiments demonstrate that our proposed module can boost the generalization and outperform other approaches, including class-balanced re-sampling, decoupled classifier re-training, and data augmentation methods. The source code is available at `https://www.lamda.nju.edu.cn/code_CSA.ashx`.

## 1 Introduction

Deep neural networks have achieved great success by applying well-designed models on large-scale elaborated datasets [1, 2, 3]. However, real-world data often exhibits a long-tail class distribution [4, 5, 6]. Learning from long-tail datasets has two main challenges, one is the class-imbalanced problem which causes the model biased towards the dominated head classes, and another is the data scarcity problem leading to the poor generalization on those rare tail classes [7, 8, 9, 10].

One simple and intuitive approach to deal with the class-imbalanced problem is re-sampling [11, 12], i.e., create the replicate of the dataset to estimate the model parameters. Unfortunately, it has been reported that re-sampling methods achieve limited effects when applied to most long-tail datasets [6, 13]. Currently, there are still few concrete and comprehensive explanations for this observation. Existing works mainly conclude that re-sampling will lead to the overfitting problem, and thus will be harmful to long-tail representation learning [6, 14].

Recently, many two-stage approaches have been proposed to improve the tail-class performance by adopting re-sampling in the second training stage. For instance, DRS [15] adopts a re-sampling

---
*equal contribution
†corresponding author

37th Conference on Neural Information Processing Systems (NeurIPS 2023).

schedule at the last several episodes of the training process. cRT [6] first trains a preliminary model using the uniform sampler, then fixes the representations and re-trains the linear classifier using a class-balanced sampler. Last but not least, BBN [14] adjusts the whole model to first learn from the conventional learning branch and dynamically move to the re-balancing branch. Overall, the two-stage method has attracted widespread attention due to its basic hypothesis that uniform sampling is beneficial to representation learning, and the class-balanced sampling can be used to fine-tune the linear classifier. In light of the success of the two-stage method, a natural question is:

*Can re-sampling benefit long-tail learning in the single-stage framework?*

To answer this question, this paper empirically studies the re-sampling strategies and finds that re-sampling leads to opposite effects on long-tail datasets. Figure 1 gives a brief view of this phenomenon. Moreover, we deduce that if the training samples are highly semantically related to their target labels, class-balanced re-sampling can learn discriminative feature representations; otherwise, uniform sampling is even better than class-balanced re-sampling, as the latter suffers from oversampling redundant unrelated contexts. To verify this, we design a pair of synthetic benchmarks with the same content but different contexts, one containing irrelevant context in training samples and the other not. Experiments confirm that re-sampling achieves conspicuous different performances on these two benchmarks. In particular, when irrelevant context exists, class-balanced re-sampling learns poorer representations compared to uniform sampling, thus the irrelevant context negatively affects re-sampling methods.

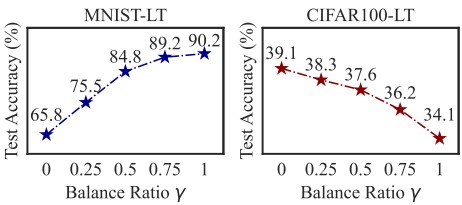

Figure 1: Performance of re-sampling on two long-tail datasets. The sampling weight for a data point $(\boldsymbol{x}, y)$ is defined as $n_y^{-\gamma}$ where $n_y$ denotes the class frequency of class $y$.

We believe that re-sampling can benefit long-tail learning in the single-stage framework. It fails on some long-tail datasets mainly because it overfits the oversampled irrelevant contexts and learns unexpected spurious correlations. If such spurious correlations are avoided, re-sampling can be helpful for long-tail learning. Motivated by this, we propose a new *context-shift augmentation* module, which transfers well-separated context from head-class data to tail-class data. Specifically, it extracts the unrelated contexts (e.g. the backgrounds or the unrelated foreground objects) from head-class images and pastes them onto tail-class images to generate diverse novel samples. In this way, it encourages the model to learn more discriminative results for the tail classes. We conduct experiments on three long-tail datasets, CIFAR10-LT, CIFAR100-LT, and ImageNet-LT. The results show that the proposed module achieves competitive performance compared to the baseline methods. In summary, our main contributions are:

- We conduct empirical analyses on different datasets and discover that re-sampling does not necessarily work or fail in long-tail learning.

- We deduce that the failure of re-sampling may be attributed to overfitting on irrelevant contexts, and our empirical studies confirm our hypothesis.

- We propose a new context-shift augmentation module to prevent re-sampling from overfitting to irrelevant contexts in a single-stage framework.

- Extensive experiments verify the effectiveness of the proposed module against class-balanced re-sampling, decoupled classifier re-training, and data augmentation methods.

The rest of the paper is organized as follows. Section 2 studies the effects of re-sampling approaches. Section 3 presents the proposed context-shift augmentation module. Section 4 briefly reviews related works. Section 5 concludes the paper.

## 2 A Closer Look at Re-sampling

### 2.1 Preliminaries

Given a training dataset $\mathcal{D} = \{\boldsymbol{x}_i, y_i\}_{i=1}^N$, where $\boldsymbol{x}_i$ is a training sample and $y_i \in \mathcal{C} = [K] = \{1, \ldots, K\}$ is the class label assigned to it. We assume that the training data follow a long-tail class

Table 1: Test accuracy (%) of CE with uniform sampling, classifier re-training (cRT), and class-balanced re-sampling (CB-RS) on four long-tail benchmarks. We report the accuracy in terms of all, many-shot, medium-shot, and few-shot classes.

| | MNIST-LT | | | | Fashion-LT | | | | CIFAR100-LT | | | | ImageNet-LT | | | |
|---|---|---|---|---|---|---|---|---|---|---|---|---|---|---|---|---|
| | All | Many | Med. | Few | All | Many | Med. | Few | All | Many | Med. | Few | All | Many | Med. | Few |
| CE | 65.8 | **99.1** | 89.9 | 0.0 | 45.6 | **94.7** | 43.1 | 0.0 | 39.1 | **65.8** | 36.8 | 8.8 | 35.0 | **57.7** | 26.5 | 4.7 |
| cRT | 82.5 | 96.6 | 89.4 | 58.8 | 60.3 | 77.1 | 61.4 | 42.1 | **41.6** | 63.0 | **40.4** | **16.5** | **41.9** | 52.9 | **39.2** | **23.6** |
| CB-RS | **90.8** | 98.7 | **94.4** | **77.7** | **80.5** | 86.6 | **74.3** | **82.8** | 34.1 | 59.5 | 31.1 | 6.2 | 37.6 | 47.5 | 36.5 | 16.7 |

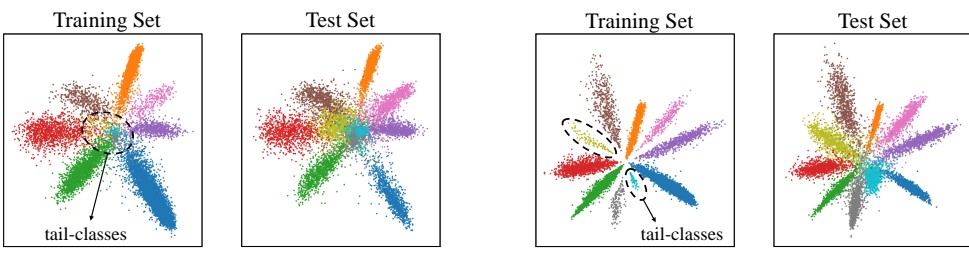

(a) Uniform sampling.          (b) Class-balanced re-sampling.

Figure 2: Visualization of learned representation of training and test set on MNIST-LT. Using class-balanced re-sampling yields more discriminative and balanced representations.

distribution where the class prior distribution $\mathbb{P}(y)$ is highly skewed so that many underrepresented classes have a very low probability of occurrence. Specifically, we define the imbalance ratio as $\rho = \max_y \mathbb{P}(y)/\min_y \mathbb{P}(y)$ to indicate the skewness of data. Classes with high $\mathbb{P}(y)$ are referred to as *head classes*, while others are referred to as *tail classes*.

In practice, since the data distribution is unknown, Empirical Risk Minimization (ERM) uses the training data to achieve an empirical estimate of the underlying data distribution. Typically, one minimizes the softmax cross-entropy as following

$$\ell(y, f(\boldsymbol{x})) = -\log \frac{\exp(f_y(\boldsymbol{x}))}{\sum_{y' \in [K]} \exp(f_{y'}(\boldsymbol{x}))} \tag{1}$$

where $f_y(\boldsymbol{x})$ denotes the predictive logit of model $f$ on class $y$. However, this ubiquitous approach neglects the issue of class imbalance and makes the model biased toward head classes [8, 16].

To deal with the class-imbalance problem, the re-sampling strategy assigns a probability of being selected for each training sample according to its class frequency [6]. The probability of sampling a data point from class $k$ can be written as:

$$p_k = \frac{n_k^q}{\sum_{k' \in [K]} n_{k'}^q} \tag{2}$$

where $n_k$ denotes the frequency of class $k$ and $q \in [0, 1]$. When $q = 1$, Equation (2) denotes uniform sampling, where each training sample has an equal probability of being selected. When $q = 0$, Equation (2) denotes the class-balanced re-sampling, which selects samples from every class $k$ with the identical probability of $1/K$.

## 2.2 Exploring the Effect of Re-sampling

### 2.2.1 Re-sampling can learn discriminative representations

To better explore the effect of the re-sampling strategy, we conduct experiments on multiple long-tail datasets, including MNIST-LT, Fashion-LT, CIFAR100-LT [15], and ImageNet-LT [5]. We compare three different learning methods: 1) Cross-Entropy (CE) with uniform sampling; 2) Classifier Re-Training (cRT) which uses uniform sampling to learn the representation and class-balanced re-sampling to fine-tune the classifier; 3) Class-Balanced Re-Sampling (CB-RS) for the whole training process. We report the experimental results in Table 1.

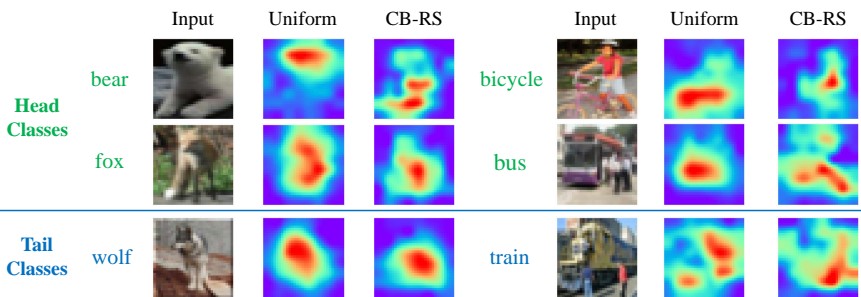

Figure 3: Visualization of features with Grad-CAM [17] on CIFAR100-LT. Uniform sampling mainly learns label-relevant features, while re-sampling overfits the label-irrelevant features.

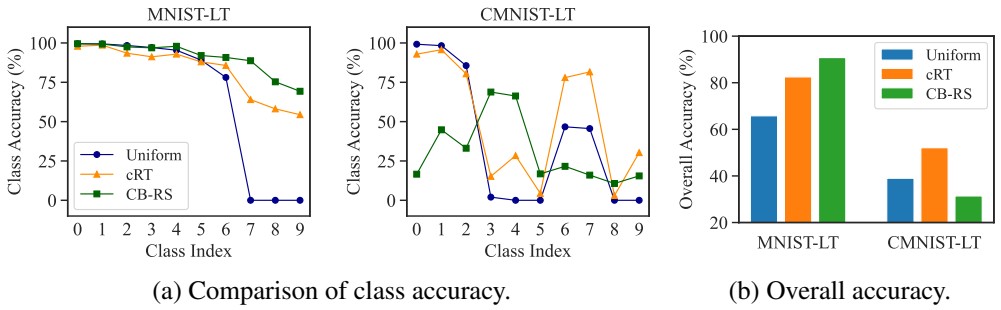

(a) Comparison of class accuracy.

(b) Overall accuracy.

Figure 4: Comparison of Uniform sampling, cRT, and CB-RS on MNIST-LT and CMNIST-LT.

According to the results, cRT performs best on CIFAR100-LT and ImageNet-LT, which is consistent with previous works [14, 6]. CE and cRT use the same representation but cRT achieves higher performances, which indicates that re-sampling can help for classifier learning. However, on MNIST-LT and Fashion-LT, CB-RS surprisingly achieves the highest performance and outperforms CE and cRT by a large margin. Since cRT and CB-RS both use class-balanced re-sampling for classifier learning, the results indicate that CB-RS learns better representations than uniform sampling on MNIST-LT and Fashion-LT.

To further understand the effect of re-sampling, we visualize the learned representation on MNIST-LT in Figure 2. The figures show that with uniform sampling, the representation space is dominated by head classes, and the representations of tail classes are hard to distinguish. By applying class-balanced re-sampling, the representations of both head and tail classes are discriminative.

### 2.2.2 Re-sampling is sensitive to irrelevant contexts

We have demonstrated the generalization ability of the re-sampling strategy on MNIST-LT and Fashion-LT. Nevertheless, re-sampling performs unsatisfactorily on the other two datasets. Since the training samples and target labels on MNIST and Fashion are highly semantically correlated [18, 19], while samples on CIFAR and ImageNet contain complex contexts [20, 3], we hypothesize that re-sampling is sensitive to the contexts in training samples.

To support our hypothesis, we visualize the Grad-CAM of the representation learned by different sampling strategies in Figure 3. When training with a uniform sampler, models can distinguish the contexts from samples of head classes. However, when adopting a class-balanced re-sampler, the model tends to overfit the irrelevant context from the over-sampled tail data, which unexpectedly affects the representation of head classes. For example, when classifying different kinds of animals, re-sampling might focus on the posture rather than the appearance. Also, when classifying different vehicles, re-sampling is easily influenced by the human in tail-class images, and thus mistakenly focuses on the human in head-class images.

To further validate our point, we design a homogeneous benchmark of MNIST-LT termed Colored-MNIST-LT (CMNIST-LT). We inject colors into MNIST-LT to artificially construct irrelevant contexts.

Specifically, we design CMNIST-LT based on two considerations. First, head classes are prone to have rich contexts, so we inject different colors into the samples of each head class. Second, tail classes have limited contexts, so we inject an identical color into the samples of each tail class. We conduct uniform sampling, classifier re-training, and class-balanced re-sampling on MNIST-LT and CMNIST-LT. The experimental results are illustrated in Figure 4. The results show that when applied to MNIST-LT, CB-RS can boost the tail-class performance without degradation on head classes. However, for CMNIST-LT, CB-RS performs worse than uniform sampling and cRT on both head and tail classes, thus validating the negative impact of irrelevant contexts on re-sampling methods. Since re-sampling succeeds on MNIST-LT and fails on CMNIST-LT, we propose that re-sampling does not always fail, it can help for long-tail learning if avoiding the irrelevant contexts.

### 2.2.3 Proposed benchmark datasets

We follow the previous works [15, 14] to construct MNIST-LT, Fashion-LT and CIFAR100-LT and set the imbalance ratio to 100. ImageNet-LT is proposed by [5]. For MNIST-LT and Fashion-LT, we use LeNet [21] as the backbone network and add a linear embedding layer before the fully connected layer to project the representation into 2-dimensional space for better presentation. We use standard SGD with a mini-batch size of 128, an initial learning rate of 0.1 and a cosine annealing schedule to train the model for 8 epochs. When applying cRT, we retrain the last fully connected layer for 4 epochs by fixing the other layers. For CIFAR100-LT and ImageNet-LT, more details are in Section 3.3.

To construct CMNIST-LT, we follow the idea of CMNIST [22] to first randomly flip the label and then inject colors into the training samples. However, different from CMNIST which converts the MNIST to a binary classification dataset, we keep the ten classes to better simulate a long-tail class distribution. To generate flipped labels on the long-tail dataset MNIST-LT, we follow the method in [23] and set the flipping probability to $1/4$. We generate ten different colors using the seaborn[2] package. For the five head classes, we randomly inject one of these ten colors into each sample with equal probability. For the other five tail classes, we inject samples of each class with a single color.

## 3  A Simple Approach to Make Re-sampling Robust to Context-shift

### 3.1  Extracting Rich Contexts from Head-class Data

By studying the effects of re-sampling methods in different scenarios, we can draw a conclusion: when the training samples contain irrelevant contexts, simply over-sampling the tail-class samples might cause the model to unexpectedly focus on these redundant contexts, thereby resulting in the overfitting problem. However, the head classes have rich data to learn a model with good generalization ability. We naturally raise a question: can we utilize the rich contexts from head data to augment the over-sampled tail data to alleviate the negative impact of irrelevant contexts? Inspired by this motivation, we design a context-shift augmentation module by extracting the rich contexts from head-class data to enrich the over-sampled tail-class data.

To leverage the rich contexts in head-class data, we utilize a model $f^u$ trained with uniform sampling for context extraction. First, we select well-learned samples with fitting probability larger than a threshold $\delta$ to improve the extraction quality. The fitting probability for sample $\boldsymbol{x}_i$ can be calculated by the Softmax function as follows

$$p(y \mid \boldsymbol{x}_i, f^u) = \frac{\exp(\boldsymbol{z}_{i,y}^u)}{\sum_{y' \in [K]} \exp(\boldsymbol{z}_{i,y'}^u)} \tag{3}$$

where $\boldsymbol{z}_i^u$ denotes the logits predicted by $f^u$, i.e., $\boldsymbol{z}_i^u = [\boldsymbol{z}_{i,1}^u, \ldots, \boldsymbol{z}_{i,K}^u] = f^u(\boldsymbol{x}_i)$. Then, we use off-the-shelf methods such as Grad-CAM [24, 17] to extract the image contexts, which is also used in previous works regarding open-set learning and adversarial learning [25, 26]. Specifically, given an image $\boldsymbol{x}_i$, we calculate its class activation map $\text{CAM}(\boldsymbol{x}_i \mid f^u)$. Then, we inverse the map to get the background mask $\boldsymbol{M}_i$, i.e., $\boldsymbol{M}_i = 1 - \text{CAM}(\boldsymbol{x}_i \mid f^u)$. Here $\boldsymbol{M}_i$ is a matrix of the same size as $\boldsymbol{x}_i$, with values between 0 and 1. A higher value indicates that the corresponding pixel is more likely to be the background. Different from previous works that discretize the mask matrix to binary values [27, 28], we keep the floating values in the matrix to conserve more information.

---

[2]http://seaborn.pydata.org/generated/seaborn.color_palette.html

After calculating the background mask $M_i$ of image $x_i$, we paste the mask onto the original image by $M_i \odot x_i$ to obtain a background image. In this way, we separate the semantically related contents from the images and keep the rest contexts for further augmentation. Finally, the extracted contexts are pushed into a memory bank $Q$ for augmentation of re-sampled data.

For the training of the uniform module, we apply the conventional ERM algorithm. For each training sample $x_i$, we calculate its loss by

$$z_i^u = f^u(x_i) \tag{4}$$
$$\mathcal{L}_i^u = \ell^u(z_i^u, y_i) \tag{5}$$

where $\ell^u$ can be any loss function. Generally, we use the standard cross-entropy loss.

## 3.2 Balanced Re-sampling with Context-shift Augmentation

Simply adopting balanced sampling might generate many repeated samples from the tail classes and lead to the overfitting problem. Therefore, we ask for background images from the context memory bank $Q$ and paste them onto the re-sampled images. In this way, we generate more diverse novel samples by simulating each tail-class image within various contexts. Specifically, for a re-sampled training image $\tilde{x}_i$, we ask for another image $\breve{x}_i$ together with its mask $M_i$ from $Q$, and fuse it with $\tilde{x}_i$ to generate a novel sample, and calculate its training loss as follow:

$$\lambda \sim \text{Uniform}(0, 1) \tag{6}$$
$$\tilde{x}_i = \lambda M_i \odot \breve{x}_i + (1 - \lambda M_i) \odot \tilde{x}_i \tag{7}$$
$$z_i^b = f^b(\tilde{x}_i) \tag{8}$$
$$\mathcal{L}_i^b = \ell^b(z_i^b, \tilde{y}_i) \tag{9}$$

Here $\lambda$ is randomly generated between $[0, 1]$ to increase the diversity. Different from previous mixup-based methods[29, 30], our method does not change the target label, because the pasted background is not related to the semantics of any class labels.

To reduce the computational complexity, the uniform module and the balanced re-sampling module can be trained simultaneously by sharing the same feature extractor $\phi$, and training their own linear classifiers $\psi^u$ and $\psi^b$, i.e., $f^u(\cdot) = \psi^u(\phi(\cdot))$ and $f^b(\cdot) = \psi^b(\phi(\cdot))$. Since the classifier is lightweight, it does not add much additional computational overhead. Moreover, the memory bank $Q$ is designed as a first-in-first-out queue with a maximum volume of $V$ for a convenient query. After extracting the context from the uniform module, we append the $x_i$ and $M_i$ pair into the context bank $Q$. When the size of $Q$ reaches its maximized volume, the oldest contexts are pushed out. In practice, the volume size is set equal to the mini-batch size in order to minimize the overhead as well as ensure the querying requirements. Finally, the summarized loss function is

$$\mathcal{L} = \mathcal{L}^u + \mathcal{L}^b = \frac{1}{N} \sum_{i=1}^{N} \mathcal{L}_i^u + \frac{1}{N} \sum_{i=1}^{N} \mathcal{L}_i^b \tag{10}$$

Figure 5 gives a brief overview of the proposed module. The detailed training procedure is given in the supplementary material due to the page limit.

In the inference phase, only the balanced re-sampling module is used. In other words, the uniform module only serves as an assistant to provide more rich contexts for the re-sampling module during the training phase. Formally, for a test data point $x$, we obtain the prediction by $z = f^b(x)$, and then employ the Softmax function to obtain the predictive probabilities.

## 3.3 Empirical Results

We demonstrate the efficacy of the proposed module *context shift augmentation* by comparing it with different kinds of long-tail learning methods, including:

- Re-sampling or re-weighting methods, such as Focal Loss [31], CB-Focal [7], CE-DRS [15], CE-DRW [15], LDAM-DRW [15], cRT [6], LWS [6], and BBN [14],
- Head-to-tail knowledge transfer methods, such as M2m [32], OLTR [5], and FSA [28],

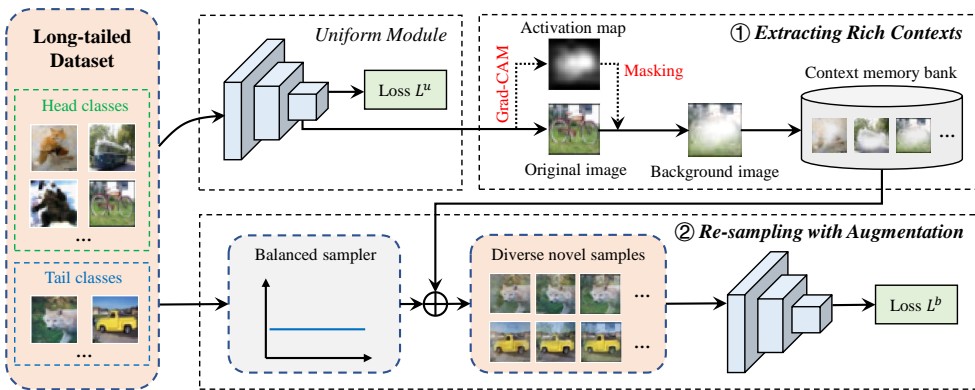

Figure 5: An overview of the proposed method.

- Data augmentation methods, such as Mixup [29], Remix [33], CAM-BS [13], and CMO [27].

We conduct experiments on three long-tail datasets, including CIFAR10-LT [15], CIFAR100-LT [15], and ImageNet-LT [5]. CIFAR10-LT and CIFAR100-LT are the long-tail versions of CIFAR datasets by sampling from the raw dataset with an imbalance ratio $\rho$. Following previous works [15, 14], we conduct experiments with $\rho \in \{100, 50, 10\}$. ImageNet-LT is a long-tail version of the ImageNet [3], which contains 1000 classes, each with a number of samples ranging from 5 to 1280. For each long-tail training dataset, we evaluate the model on another corresponding class-balanced test set by calculating the overall prediction accuracy.

The results for CIFAR10-LT and CIFAR100-LT are summarized in Table 2. We report the results under imbalance ratio $\rho = 100, 50, 10$. As shown in the table, our method achieves superior performance compared with the baseline methods in all settings.

Table 2: Test accuracy (%) on CIFAR datasets with various imbalanced ratios.

| Dataset | CIFAR100-LT | | | CIFAR10-LT | | |
|---|---|---|---|---|---|---|
| Imbalance Ratio | 100 | 50 | 10 | 100 | 50 | 10 |
| CE | 38.3 | 43.9 | 55.7 | 70.4 | 74.8 | 86.4 |
| Focal Loss [31] | 38.4 | 44.3 | 55.8 | 70.4 | 76.7 | 86.7 |
| CB-Focal [7] | 39.6 | 45.2 | 58.0 | 74.6 | 79.3 | 87.1 |
| CE-DRS [15] | 41.6 | 45.5 | 58.1 | 75.6 | 79.8 | 87.4 |
| CE-DRW [15] | 41.5 | 45.3 | 58.1 | 76.3 | 80.0 | 87.6 |
| LDAM-DRW [15] | 42.0 | 46.6 | 58.7 | 77.0 | 81.0 | 88.2 |
| cRT [6] | 42.3 | 46.8 | 58.1 | 75.7 | 80.4 | 88.3 |
| LWS [6] | 42.3 | 46.4 | 58.1 | 73.0 | 78.5 | 87.7 |
| BBN [14] | 42.6 | 47.0 | 59.1 | 79.8 | 82.2 | 88.3 |
| mixup [29] | 39.5 | 45.0 | 58.0 | 73.1 | 77.8 | 87.1 |
| Remix [33] | 41.9 | - | 59.4 | 75.4 | - | 88.2 |
| M2m [32] | 43.5 | - | 57.6 | 79.1 | - | 87.5 |
| CAM-BS [13] | 41.7 | 46.0 | - | 75.4 | 81.4 | - |
| CMO [27] | 43.9 | 48.3 | 59.5 | - | - | - |
| cRT+mixup [34] | 45.1 | 50.9 | 62.1 | 79.1 | 84.2 | 89.8 |
| LWS+mixup [34] | 44.2 | 50.7 | 62.3 | 76.3 | 82.6 | 89.6 |
| CSA (ours) | 45.8 | 49.6 | 61.3 | 80.6 | 84.3 | 89.8 |
| CSA + mixup (ours) | **46.6** | **51.9** | **62.6** | **82.5** | **86.0** | **90.8** |

Specifically, the methods based on re-sampling or re-weighting such as DRS, DRW, cRT, and LWS can ease the class-imbalanced problem to some degree but the performance gain is limited due to the neglect of the representation learning. The methods based on knowledge transfer and data augmentations, such as M2m and CMO, achieve higher performance.

Moreover, by combining cRT and LWS with mixup, the performance achieves an obvious improvement. However, such two-stage training methods are not end-to-end approaches. In contrast, the proposed context shift augmentation module enhances representation learning by enriching the contexts of samples and adopting the class-balanced re-sampling to ensure a balanced classifier. In this manner, it achieves more improvement with an end-to-end framework. Since cRT and the proposed module both use class-balanced sampling for classifier learning, the results indicate that the proposed context shift augmentation can achieve better representations.

We further conduct the experiments on a larger scale dataset ImageNet-LT. We calculate the accuracy of the overall test set and the average accuracy of the many-shot classes (more than 100 images in the training set), the medium-shot ($20\sim100$ images), and the few-shot classes (less than 20 images). We report the results in Table 3. It shows that the proposed module is superior to most other baseline methods. The performance is similar to another data augmentation method CMO, but our method achieves higher accuracy on few-shot classes, which demonstrates the generalization ability of context-shift augmentation for tail-class data.

Table 3: Test accuracy (%) on ImageNet-LT dataset.

| | ResNet-10 (All) | ResNet-50 | | | |
| --- | --- | --- | --- | --- | --- |
| | | All | Many | Med. | Few |
| CE | 34.8 | 41.6 | 64.0 | 33.8 | 5.8 |
| Focal Loss [31] | 30.5 | - | - | - | - |
| OLTR [5] | 35.6 | - | - | - | - |
| FSA [28] | 35.2 | - | - | - | - |
| cRT [6] | 41.8 | 47.3 | 58.8 | 44.0 | 26.1 |
| LWS [6] | 41.4 | 47.7 | 57.1 | 45.2 | 29.3 |
| BBN [14] | - | 48.3 | - | - | - |
| CMO [27][†] | - | 49.1 | 67.0 | 42.3 | 20.5 |
| CSA (ours) | 42.7 | 49.1 | 62.5 | 46.6 | 24.1 |
| CSA[†] (ours) | **43.2** | **49.7** | 63.6 | 47.0 | 23.8 |

[†] denotes a longer training of 100 epochs.

We provide more detailed studies to analyze the effect of each component in the proposed module. Due to the page limit, we report the results in the supplementary material.

## 4 Related Work

### 4.1 Re-sampling and Re-weighting

Re-sampling is a widely used strategy in class-imbalanced learning [11, 12]. There are two main ideas of re-sampling: 1) Over-sampling by repeating data from the rare classes. 2) Under-sampling by abandoning a proportion of data from the frequent classes. However, when the class distribution is highly skewed, re-sampling methods often fail. Previous works point out that under-sampling may discard precious information which inevitably degrades the model performance, and over-sampling tends to cause the overfitting problem on the tail classes [14, 15].

Recent work [6] develops class-balanced re-sampling where samples from each class have an identical probability of being sampled. Class-balance re-sampling can bring performance gain for classifier learning but hurts representation learning. Therefore, two-stage approaches [15, 6, 14] adopt it at the late stage of the whole training process in order not to impact the representation.

Re-weighting aims to generate more balanced predictions by adjusting the losses for different classes [35, 36]. The most intuitive way is to weight each training sample by the inverse of its class frequency [4]. Similar to class-balanced re-sampling, re-weighting can achieve better results for tail classes but usually deteriorates its performance for head classes [7].

In contrast, our empirical study reveals that class-balanced re-sampling can be an effective method as long as there exist no irrelevant contexts. It fails in some cases mainly due to the unexpected overfitting

towards the over-sampled redundant contexts. When applied with context-shift augmentation, class-balanced re-sampling can achieve competitive performance on long-tail datasets.

## 4.2 Head-to-tail Knowledge Transfer

As the head classes have adequate training data while the tail classes have limited data, recent works aim to leverage the knowledge gained from head classes to enhance the generalization of tail classes. Feature transfer learning [37] utilizes the intra-class variance from head classes to guide the feature augmentation for tail classes. MetaModelNet [4] uses the head data to train a meta-network to predict many-shot model parameters from few-shot model parameters, then transfers the meta-knowledge to the tail classes. Major-to-minor translation (M2m) [32] uses the over-sampling method and translates the head-class samples to replace the duplicated tail-class samples via adversarial perturbations. OLTR [5] maintains a dynamic meta-embedding between head and tail classes to transfer the semantic deep features from head to tail classes.

These methods assume that the head classes and the tail classes share some common knowledge such as the same intra-class variances, the same model parameters, or the same semantic features. In this work, we regard the contexts as such knowledge and transfer the contexts from head-class data to enrich the tail-class data.

## 4.3 Data Augmentation

Several data augmentation approaches have been proposed to improve the model generalization ability. In contrastive learning [38, 39], curriculum learning [40], meta-learning methods [41] and instance segmentation tasks [42], data augmentation strategies have been shown to effectively improve the generalization of tail classes. MiSLAS [34] studies the mixup [29] technology in long-tail learning and finds that mixup can have a positive effect on representation learning but a negative or negligible effect on classifier learning. Remix [33] adapts the mixup method to a re-balanced version. It assigns the mixed label in favor of the tail class by designing a disproportionately higher weight for the tail class. CMO [27] applies CutMix [30] by cutting out random regions of a sample from head classes and filling the removed regions with another sample from tail classes. By this means, it enriches the contexts of the tail data; but the random cutout operation does not necessarily separate the contents and contexts.

The attention or CAM-based methods have been proposed to improve long-tail learning via feature decomposition and augmentation. CAM-BS [13] separates the foreground and background of each sample, then augments the foreground part by flipping, translating, rotating, or scaling. Feature Space Augmentation (FSA) [28] uses CAM to decompose the features of each class into a class-generic component and a class-specific component, and generates novel samples in the feature space by combining the class-specific components from the tail classes and the class-generic components from the head classes. Attentive Feature Augmentation (AFA) [43] adopts feature decomposition and augmentation via the attention mechanism. Note that FSA and AFA can also be seen as head-to-tail knowledge transfer approaches. Nevertheless, these methods neglect that the learned model has limited generalization ability on tail classes, and most foregrounds (or class-specific components) of samples from tail classes are incredible. In comparison, our method applies CAM to separate related contents and unrelated contexts for samples mainly from head classes. It then pastes the contexts extracted from the head-class data onto the over-sampled tail-class data to enrich the contexts.

## 5 Conclusion

In this work, we study the re-sampling strategy for long-tail learning. Our empirical investigations reveal that the impact of re-sampling is highly dependent on the existence of irrelevant contexts and is not always harmful to long-tail learning. To reduce the influence of irrelevant contexts, we propose a new context-shift augmentation module that leverages the well-separated contexts from the head-class images to augment the over-sampled tail-class images. We demonstrate the superiority of the proposed module by conducting experiments on several long-tail datasets and comparing it against class-balanced re-sampling, decoupled classifier re-training, and data augmentation methods.

## Broader Impact and Limitations

This paper investigates the reasons behind the success/failure of re-sampling approaches in long-tail learning. In critical and high-stakes applications, such as medical image diagnosis and autonomous driving, the presence of imbalanced data poses the risk of producing biased predictions. By shedding light on this problem, we aim to inspire more research on safe and robust re-sampling approaches.

One may be concerned about combining the proposed module with other methods such as self-supervised learning [44], logit adjustment [16]. We conduct additional experiments and report the results in the supplementary material due to the page limit. Nevertheless, the proposed module can not achieve comparable performance with the well-designed models [45, 46, 47], since our intention is not to achieve performance that is on par with state-of-the-art methods. Instead, we hope that our findings will inspire future research regarding re-sampling methods.

## Data Availability Statement

The source code of our method is available at `https://www.lamda.nju.edu.cn/code_CSA.ashx` or `https://github.com/shijxcs/CSA`.

## Acknowledgments and Disclosure of Funding

This research was supported by the National Key R&D Program of China (2022ZD0114803), the National Science Foundation of China (62176118, 61921006).

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

# A   Training Procedure

The training procedure of *context-shift augmentation* is summarized in Algorithm 1.

---

**Algorithm 1** Training procedure of context-shift augmentation

---

**Input:** training data $\mathcal{D} = \{(\boldsymbol{x}_i, y_i)\}_{i=1}^N$; context memory bank $Q$, maximum volume size $V$; model parameters $\phi$, $f^u$, $f^b$; loss functions $\ell^u$, $\ell^b$;

**Procedure:**

1: Initialize model parameters $\phi$, $f^u$, $f^b$;
2: Re-sampling a class-balanced dataset $\widetilde{\mathcal{D}} = \{(\tilde{\boldsymbol{x}}_i, \tilde{y}_i)\}_{i=1}^N$;
3: Empty memory bank $Q$;
4: **for** epoch $= 1, \ldots, T$ **do**
5:    **repeat**
6:       Draw a mini-batch $(\boldsymbol{x}_i, y_i)_{i=1}^B$ from $\mathcal{D}$;
7:       Draw a mini-batch $(\tilde{\boldsymbol{x}}_i, \tilde{y}_i)_{i=1}^B$ from $\widetilde{\mathcal{D}}$;
8:       *// uniform module*
9:       **for** $i = 1, \ldots, B$ **do**
10:          Calculate $\boldsymbol{z}_i^u = f^u(\phi(\boldsymbol{x}_i))$ and $\mathcal{L}_i^u = \ell^u(\boldsymbol{z}_i^u, y_i)$;
11:          **if** $p(y = y_i \mid \boldsymbol{x}_i, \phi, f^u) \geq \delta$ **then**
12:             Calculate background mask $\boldsymbol{M}_i$ of $\boldsymbol{x}_i$;
13:             Push $(\boldsymbol{x}_i, \boldsymbol{M}_i)$ into $Q$;
14:          **end if**
15:       **end for**
16:       Calculate $\mathcal{L}^u = \frac{1}{B} \sum_{i=1}^B \mathcal{L}_i^u$;
17:       *// balanced re-sampling module*
18:       **if** Size of $Q$ reaches $V$ **then**
19:          Obtain contexts $(\breve{\boldsymbol{x}}_i, \boldsymbol{M}_i)_{i=1}^B$ from $Q$;
20:          $\lambda \sim \text{Uniform}(0, 1)$;
21:          **for** $i = 1, \ldots, B$ **do**
22:             $\tilde{\boldsymbol{x}}_i = \lambda \boldsymbol{M}_i \odot \breve{\boldsymbol{x}}_i + (1 - \lambda \boldsymbol{M}_i) \odot \tilde{\boldsymbol{x}}_i$;
23:             Calculate $\boldsymbol{z}_i^b = f^b(\phi(\tilde{\boldsymbol{x}}_i))$ and $\mathcal{L}_i^b = \ell^b(\boldsymbol{z}_i^b, \tilde{y}_i)$;
24:          **end for**
25:          Calculate $\mathcal{L}^b = \frac{1}{B} \sum_{i=1}^B \mathcal{L}_i^b$;
26:       **else**
27:          Assign $\mathcal{L}^b = 0$;
28:       **end if**
29:       *// total objective function*
30:       Calculate $\mathcal{L} = \mathcal{L}^u + \mathcal{L}^b$;
31:       Update model parameters $\phi$, $f^u$, $f^b$ with $\mathcal{L}$;
32:    **until** all training data are traversed.
33: **end for**

---

# B   Implementation Details for *context-shift augmentation*

For experiments on CIFAR10-LT and CIFAR100-LT, we use ResNet-32 as the backbone network and train it using standard SGD with a momentum of 0.9, a weight decay of $2 \times 10^{-4}$, a batch size of 128. The model is trained for 200 epochs. The initial learning rate is set to 0.2 and is annealed by a factor of 10 at 160 and 180 epochs. We train each model with 1 NVIDIA GeForce RTX 3090.

For experiments on ImageNet-LT, we implement the proposed method on ResNet-10 and ResNet-50. We use standard SGD with a momentum of 0.9, a weight decay of $5 \times 10^{-4}$, and a batch size of 256 to train the whole model for a total of 90 epochs. We use the cosine learning rate decay with an initial learning rate of 0.2. We train each model with 2 NVIDIA Tesla V100 GPUs.

In all experiments, we first warm up the uniform module for 10 epochs and then train the uniform module and the balanced re-sampling module simultaneously for the rest epochs. For the uniform module, we follow the simple data augmentation used in [2] with only random crop and horizontal

flips. For the re-sampling module, we use the proposed context-shift augmentation method. We apply the trick proposed by [48] to disable the augmentation in the balanced re-sampling module at the last 3 epochs to obtain further improvements, which is also applied in other baseline methods [13, 27]. We set the threshold $\delta$ to 0.8.

## C  Additional Illustrations

For convenience in understanding context and content, we give an example in Figure 6. Context refers to the semantically unrelated parts in the images, and content refers to the semantically related parts. Moreover, We give a brief illustration of the generated dataset CMNIST-LT in Figure 7.

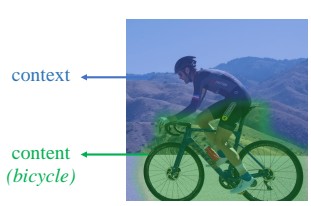

Figure 6: Illustration of context and content. Taking a photo of the bicycle as an example.

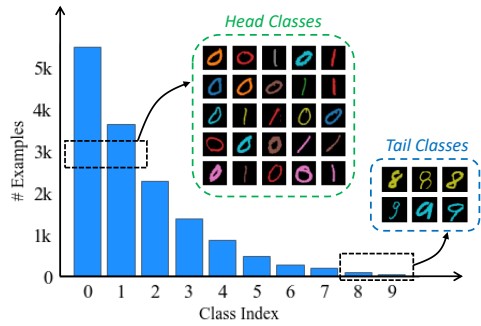

Figure 7: Illustration of the CMNIST-LT benchmark.

## D  Additional Experimental Results

### D.1  Effects of the context bank

The context bank $Q$ is a novel component of *context-shift augmentation* which receives diverse contexts from the uniform module and provides them to augment the data in the re-sampling module. To verify the effectiveness of the context bank, we remove it from the framework and train the model on CIFAR100-LT with an imbalance ratio of 100. The results are reported in Table 4.

Table 4: Ablation study on the context bank $Q$.

|            | All    | Many   | Med.   | Few    |
|------------|--------|--------|--------|--------|
| Ours       | 45.8   | 64.3   | 49.7   | 18.2   |
| Ours w/o $Q$ | 41.2 (-4.6) | 65.1 (+0.8) | 41.9 (-7.8) | 10.7 (-7.5) |

The results show that without context bank $Q$, the performance decreases by a large margin. The performance degradation mostly comes from the medium-shot classes and the few-shot classes, which indicates that the context bank can significantly improve the generalization of tail classes.

Moreover, we study the effect of different variants in the context bank. First, we study the threshold $\delta$ for sample selection and report the results in Table 5. On the one hand, if $\delta$ is too high, the selected samples will be very few. On the other hand, if $\delta$ is too small, the selected samples might be not well learned. Nevertheless, as the training process progresses, most samples will fit well, so our method is not sensitive to $\delta$. We set $\delta = 0.8$ considering its best performance.

Second, we study the influence of the volume size $V$ of the context bank $Q$ and report the results in Table 6. Since the bank $Q$ is a first-in-first-out queue, the latest incoming contexts are more convincing. When the volume size is too large, the bank might contain more past samples. Besides, a larger size of $V$ would bring more memory overhead. So we set the volume size $V$ equal to the mini-batch size $B$ in our method.

Table 5: Influence of the threshold $\delta$.

| $\delta$ | 0 | 0.1 | 0.2 | 0.3 | 0.4 | 0.5 | 0.6 | 0.7 | 0.8 | 0.9 |
|---|---|---|---|---|---|---|---|---|---|---|
| Accuracy | 45.47 | 45.37 | 45.55 | 44.83 | 45.52 | 45.59 | 45.42 | 45.08 | **45.83** | 44.93 |

Table 6: Influence of the bank volume size (compared with the mini-batch size $B$).

| Volume | $\times 1$ | $\times 2$ | $\times 4$ | $\times 8$ | $\times 16$ | $\times 32$ | $\times 64$ |
|---|---|---|---|---|---|---|---|
| Accuracy | **45.83** | 45.60 | 45.57 | 45.32 | 45.55 | 45.37 | 45.26 |

## D.2 Influence of augmentation variants

We use a variant $\lambda \sim \mathrm{Uniform}(0, 1)$ for generating novel samples. The value of $\lambda$ can result in different proportions of foreground and background in the novel sample. Also, the size of the sampling space affects the diversity of the novel images. To explore the effect of $\lambda$, we try $\lambda \sim \mathrm{Uniform}(a, b)$ and $\lambda \sim \mathrm{Beta}(a, b)$ to train *context-shift augmentation* on CIFAR100-LT with imbalance ratio 100 and report the results in Figure 8.

First, when $\lambda$ is close to 0, the background merely takes effect, and the performance decreases a lot. Second, when $\lambda = 1$, the background image might cover the important content in the foreground image. Also, the diversity of new samples is limited. Although the performance is better than that of $\lambda = 0$, it is still unsatisfactory. Overall, choosing $\lambda \sim \mathrm{Uniform}(0, 1)$ or $\lambda \sim \mathrm{Beta}(1, 1)$ lead to the best performance.

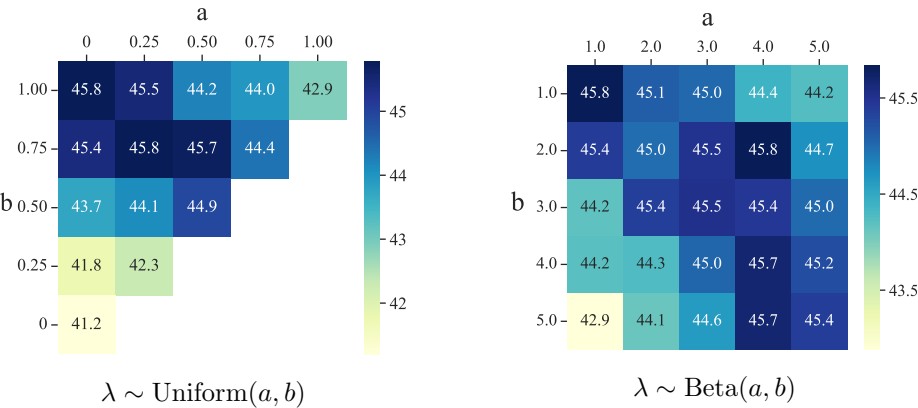

Figure 8: Influence of variants $\lambda$.

## D.3 Comparison between different modules

In our framework, the uniform sampling module is only enabled in the training phase. While in the inference phase, we use the balanced re-sampling module to predict unseen instances. To verify the superiority of the re-sampling module, we compare the performance of these two modules as well as their ensemble. We report the results in Table 7. The results show that the re-sampling module is superior to the uniform module, and even achieves higher accuracy than the ensembled results. This also indicates the superiority of the proposed context-shift re-sampling method.

Moreover, we study the influence of different balance ratios on our re-sampling module and compare it with the vanilla re-sampling method. We report the results in Figure 9. For vanilla re-sampling, adopting a more balanced re-sampling would yield more severe performance degradation. In contrast, our method achieves higher performance through class-balanced resampling.

Table 7: Comparison between the uniform module and the re-sampling module in *context-shift augmentation*.

|  | All | Many | Med. | Few |
|---|---|---|---|---|
| Uniform module | 39.4 | **68.3** | 37.3 | 6.1 |
| Re-sampling module | **45.8** | 64.3 | **49.7** | **18.2** |
| Ensemble results | 43.0 | 67.5 | 44.1 | 11.4 |

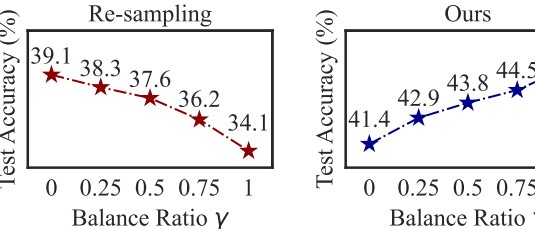

Figure 9: Comparison of re-sampling and our method under different balance ratios $\gamma$.

## D.4 Comparison of the learned representation

We visualize the representation of CIFAR100-LT in Figure 10. Each color represents a class, with darker colors representing head classes, and lighter colors representing tail classes. Although the fine-grained colors make it difficult to distinguish some classes, it can still be seen that CB-RS learns worse representation compared with vanilla CE. Moreover, our proposed method can learn a more distinguishable representation. Moreover, we visualize the Grad-CAM for examples with our proposed context-shift augmentation. We choose the same examples in Figure 3 in the main paper and report the results in Figure 11. The results show that our method can alleviate the negative impact on head-class samples caused by the overfitting problem.

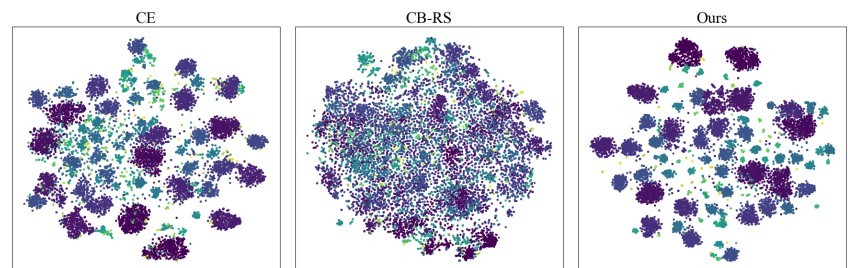

Figure 10: Visualization of learned representation on CIFAR100-LT.

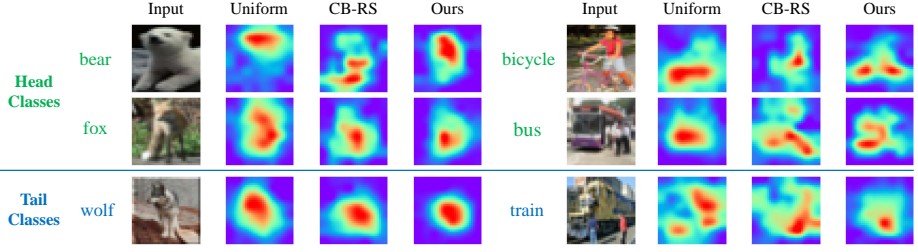

Figure 11: Visualization of features with Grad-CAM on CIFAR100-LT. Our method can alleviate the negative impact on head-class samples caused by the overfitting problem.

## D.5 The influence of Grad-CAM

The Grad-CAM [24] is utilized to extract unrelated contexts in previous works such as open-set learning and adversarial learning [25, 26]. Also, we use Grad-CAM in the *context-shift augmentation* to generate diverse contexts for tail-class data. We compare Grad-CAM with CAM [17]. The results shown in Tab. 8 demonstrate that Grad-CAM is superior to CAM when applied to our method. One may be concerned with the generated activation map of tail-class images. We visualize some tail-class samples with predicted probabilities higher than $\delta$ in Figure 12, which shows that the model can still grap accurate activation maps for tail-class samples.

Table 8: Comparison between CAM and Grad-CAM in *context-shift augmentation*

|                  | All      | Many | Med.     | Few      |
|------------------|----------|------|----------|----------|
| Ours w/ CAM      | 45.1     | 63.8 | 48.1     | 18.0     |
| Ours w/ Grad-CAM | **45.8** | 64.3 | **49.7** | **18.2** |

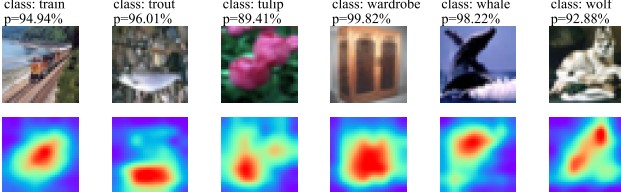

Figure 12: Visualization of features with Grad-CAM. For tail-class samples with predicted probabilities higher than the threshold (0.8), the model can also grap accurate activation maps.

## D.6 Combination with self-supervised learning

It is interesting to combine self-supervised methods with *context-shift augmentation*. Inspired by this, we follow the self-supervised + fine-tune method, i.e., SimSiam+rwSAM in [44] and conduct extensive experiments on the CIFAR10-LT dataset. The results are shown in Tab. 9.

Table 9: Combing self-supervised method SimSiam+rwSAM with different re-balancing methods

| Pre-train model | Fine-tune method | Accuracy (%) |
|-----------------|------------------|--------------|
| SimSiam+rwSAM   | CE               | 69.5         |
| SimSiam+rwSAM   | CB-RS            | 72.8         |
| SimSiam+rwSAM   | Ours             | **75.5**     |

Note that in [44], the models are pre-trained with the long-tailed dataset, while fine-tuned with the balanced in-domain dataset. However, it is hard to achieve a balanced in-domain version of a long-tailed dataset in real-world scenarios. So we use the long-tailed dataset with a balancing method including class-balanced re-sampling (CB-RS), and re-sampling with *context-shift augmentation*. The results show that our method is superior to CB-RS.

## D.7 Combination with the logit adjustment

Since our work aims to study the effectiveness of re-sampling in long-tail learning, we use the Class-Balanced Re-Sampling (CB-RS) in our method. We consider combining our method with other re-balancing methods, such as Logit Adjustment (LA) [16]. Specifically, we change the class-balanced re-sampling to the uniform sampling while adopting *context-shift augmentation*. Moreover, we consider combining class-balanced re-sampling and LA simultaneously. The comparison results are shown in Tab. 10. The results show that our method can be combined with logit adjustment to yield a higher performance. However, by applying the balanced loss and the balanced sampling, the model puts much focus on tail classes and results in a deterioration of overall accuracy.

Table 10: Combination with Logit Adjustment (LA). Accuracy results (%) on CIFAR100-LT with an imbalance ratio of 100 are reported.

|  | All | Many | Med. | Few |
|---|---|---|---|---|
| w/ CB-RS | 45.8 | **64.3** | 49.7 | 18.2 |
| w/ LA | **47.2** | 62.4 | 48.8 | 26.1 |
| w/ both | 45.0 | 48.7 | **51.0** | **31.4** |

## D.8    Combination with supervised contrastive learning

Our proposed *context-shift augmentation* (abbreviated as CSA) can be integrated with the supervised contrastive learning method BCL [39] to further improve the generalization. In Table 11, we report the experimental results. We conduct experiments on CIFAR100-LT with varying imbalance ratios, showing that CSA consistently boosts the performance of BCL.

Table 11: Accuracy (%) on CIFAR100-LT by integrating the proposed CSA into BCL

| Imbalance Ratio | 100 | 50 | 10 |
|---|---|---|---|
| BCL | 51.9 | 56.6 | 64.9 |
| BCL w/ CSA | **52.6** | **57.1** | **65.8** |

## D.9    Computational cost analysis of *context-shift augmentation*.

The proposed *context-shift augmentation* is based on the widely-used dual-branch network. Moreover, a context-extracting module is designed to calculate Grad-CAM as well as obtain the contexts. The context-extracting module has a very small computational cost, as it only needs to apply the gradient backward once at the last layer of the network, and can be ignored compared to the global gradient backward for updating the whole model. Besides, it does not spend too much memory space to save the extracted contexts, since the size of the context bank is set equal to the mini-batch size. In Table 12, we report the training time cost per epoch of CIFAR100-LT using a single RTX3090, which also demonstrates that the proposed method does not lead to too much additional computational overhead.

Table 12: Training time cost per epoch on CIFAR100-LT.

|  | w/ CE | w/ BCL |
|---|---|---|
| Single-Branch | 2.04 s | 4.76 s |
| Dual-Branch | 2.38 s | 4.85 s |
| Ours | 2.98 s | 5.10 s |

