# OpenReview forum: "How Re-sampling Helps for Long-Tail Learning?"
_NeurIPS.cc/2023/Conference — NeurIPS 2023 poster_

### Official Review · Reviewer_J7TL · 2023-06-21

**Soundness:** 1 poor
**Presentation:** 2 fair
**Contribution:** 1 poor
**Rating:** 5
**Confidence:** 4

**Summary:**

The authors suggest looking closer at the resampling strategy for long-tail recognition. The motivation comes from the observation that in recent works resampling brings rather negative than positive effects. Based on MNIST experiments, the authors demonstrate that context (background) imposes spurious correlations during training that prevents generalization on tail classes. To deal with such correlations, the authors suggest a context shift augmentation module to generate diverse training images for the tail classes. Experimental results show improvement over listed by the authors baselines.

**Strengths:**

S1: The authors suggest an interesting direction of analysis, namely why re-sampling fails to help generalization of the tail classes. The study is performed on the toy dataset with the aim to demonstrate that spurious correlations change drastically the predictions.

S2: The overall structure and the storyline are nice; it is easy to follow the method section

**Weaknesses:**

W1: The synthetic experiment does not support the statements of the authors. The claim that spurious correlations hinder tail class accuracy is not presented in Fig.4 (a) on CMNIST-LT. Namely, starting from class 3, all the accuracies are jumping around independently of the number of samples in the class and the sampling strategy. “Overfit to the irrelevant context from the over-sampled tail data” is also not observed. Moreover, the generated dataset is not visualized. Further, the statement on L162-L165 sounds reasonable; however, I find the provided arguments not convincing and rather unrelated.

W2: the authors claim to improve generalization on the tail classes; however, based on Table 3 the improvement comes mostly from “many” and “med” classes.

W3: the tables overall are misleading. The authors simply do not include multiple baselines that outperform their method such as
- Long-Tail Learning via Logit Adjustment
- Parametric Contrastive Learning
- Balanced Contrastive Learning for Long-Tailed Visual Recognition

W4: The authors briefly discuss differences to [28], but it looks like the novelty is rather limited wrt this method.

W5: the presentation of the paper can be improved. Some sentences are hard to parse and connect to the previous context without reading further, e.g. L10, L44, L51-52. The caption of Fig.1 does not help to understand Fig.1, related text in the intro also is not helpful.

**Questions:**

Q1: Would it be possible to conduct a similar experiment on more complex data with a more in-depth analysis of the correlations? It would significantly help to improve the paper.

Q2: Is there any adaptation/analysis of the spatial structure of the images, e.g. if the object is not in the center of the image or there are multiple objects of interest?

Q3: in Eq. (7), the mask M relates to image x_double_hat but is unrelated to x_tilde. In this case not only background is affected by the augmentation. I didn’t find the discussion in this regard.

Q4: Could the authors discuss more in detail the difference to [28] and provide quantitative confirmation of the improvements based purely on the proposed novelty?

**Limitations:**

The authors mention that their method does not achieve comparable performance to papers presented at CVPR/ICCV21. No further discussion is added.
Other limitation in regard medical images and autonomous driving is rather general than specific to the method.

---

> ### Author Rebuttal · Authors · 2023-08-09
>
> We appreciate the reviewer for the thoughtful reviews, and for commenting that the overall structure and the storyline are 'nice'. We will address the concerns below.
>
> **Concern #1 (Weakness #1 & Question #1): The synthetic experiment does not support the statements. “Overfit to the irrelevant context from the over-sampled tail data” is also not observed. Moreover, the generated dataset is not visualized.**
>
> **Response:** Thank you for the concerns. There might be some misunderstandings.
>
> - The overfitting problem occurs on the tail classes, but will affect the performance of **all classes, not only the tail classes**. The reason is that re-sampling the tail-class samples will lead to overfitting to the irrelevant contexts, thus inevitably leading to misclassification from the head to the tail classes. Therefore, both the head and the tail classes in Fig 4(a) have a performance decrease, and **the main decrease comes from the head classes**. This phenomenon is also shown in Figure 3 in the main paper, where the head-class feature is affected by CB-RS more obviously. Moreover, we visualize the representation of CIFAR100-LT in Figure (r1) in the Rebuttal PDF file, which shows that CB-RS learns worse representation than CE on all classes.
> - “Overfit to the irrelevant context from the over-sampled tail data” is hard to be observed. However, the injected color is the single factor that lead to opposite results on MNIST-LT and CMNIST-LT, thus demonstrating the effects of the irrelevant contexts. Moreover, the training and test accuracy can also demonstrate the overfitting of CB-RS, as is shown below.
>
>   |       | training acc | test acc |
>   | ----- | ------------ | -------- |
>   | CE    | 60.75        | 39.07    |
>   | CB-RS | 69.40        | 31.40    |
>
> - Moreover, We give a brief illustration of the generated dataset in Figure (r5) in the Rebuttal PDF file.
>
> **Concern #2 (Weakness #2): Based on Table 3 the improvement comes mostly from “many” and “med” classes.**
>
> **Response:** Our empirical study shows that re-sampling can leads to the overfitting of irrelevant features, thus affecting the performance of head classes (See Figure 3 and explanation in Section 2.2.2). Therefore, solving this problem can alleviate the negative impact on head classes, and improve the corresponding performance. There is a trade-off between head- and tail-class performance, and our method achieves a higher overall performance, which demonstrates the generalization abilities among all classes.
>
> **Concern #3 (Weakness #3): Include more baselines.**
>
> **Response:** Thanks for your suggestion. We have combined our method with LA and reported the results in Section C.7 in the supplementary material. Following are the experimental results.
>
> |            | Overall | Many | Medium | Few  |
> | ---------- | ------- | ---- | ------ | ---- |
> | Ours       | 45.8    | 64.3 | 49.7   | 18.2 |
> | Ours w/ LA | 47.2    | 62.4 | 48.8   | 26.1 |
>
> Moreover, our proposed Context-Shift Augmentation (abbreviated as CSA) can be integrated with Contrastive-Learning based method BCL to further improve the generalization. Following are the experimental results.
>
> | CIFAR100-LT         | ir=100 | ir=50 | ir=10 |
> | ------------------- | ------ | ----- | ----- |
> | BCL                 | 51.9   | 56.6  | 64.9  |
> | BCL w/ proposed CSA | 52.6   | 57.1  | 65.8  |
>
> **Concern #4 (Weakness #4 & Question #4): The novelty is rather limited wrt FSA.**
>
> **Response:** Our work is the first that conduct systematical empirical studies to investigate the reasons behind the success/failure of re-sampling. Moreover, we propose a context-shift augmentation method. Compared to FSA, our method directly applies data augmentation to the original image, which is closer to the well-known mixup [1] method.
>
> **Concern #5 (Weakness #5): The presentation of the paper can be improved.**
>
> **Response:** Thank you for the suggestion. We will improve the presentation in the next version.
>
> **Concern #6 (Question #2): Is there any adaptation/analysis of the spatial structure of the images, e.g. if the object is not in the center of the image or there are multiple objects of interest?**
>
> **Response:** According to the paper [2], the bounding boxes of objects mainly appear in the center. Moreover, no matter whether the object is in the center of the image, we can detect the location of the semantically related parts by using Grad-CAM. And our proposed method will not change the semantical information of the augmented images regardless of the spatial structure of the images.
>
> **Concern #7 (Question #3): In Eq. (7), the mask M relates to image x_double_hat but is unrelated to x_tilde.**
>
> **Response:** The original mixup [1] method also randomly mixes the unrelated two images. The other mixup-based methods, such as CutMix [3], Remix [4], and CMO [5], also mix unrelated images. Therefore, we follow these works and mix unrelated images in our method.
>
> [1] Mixup: Beyond empirical risk minimization.
>
> [2] ImageNet: A large-scale hierarchical image database.
>
> [3] Cutmix: Regularization strategy to train strong classifiers with localizable features
>
> [4] Remix: Rebalanced mixup.
>
> [5] The majority can help the minority: Context-rich minority oversampling for long-tailed classification.

---

> > ### Comment · Reviewer_J7TL · 2023-08-18
> >
> > Dear authors,
> >
> > thank you for the clarification of my misunderstanding in regard to W1&Q1, visualisation of the dataset and providing additional results.
> > I believe that the topic that the authors cover in the current submission is important and very interesting. However, the paper lack systematic analysis on more complex data. Otherwise, the findings are not applicable to any computer vision benchmark / real world application.
> >
> > After reading thoroughly the author response and other reviews, I incline to keep my rating.
> > 1. The analysis is done on a toy dataset that exhibit only one spurious feature. Based on [1] or [2], the analysis can be done more systematically taking in account more variations in the data.
> > 2. The proposed method lacks novelty wrt [28] as the authors do not disagree in the rebuttal and rather state their main novelty in the analysis; however, see 1.
> > 3. The method do not achieve sota numbers, which in combination with 1. and 2. does not strengthen the submission.
> >
> > [1] “Can contrastive learning avoid shortcut solutions?” Robinson, Joshua, et al. NeurIPS 2021
> > [2] “Towards Better Understanding Attribution Methods” Rao S., et al, CVPR 2022

---

> > > ### Author Response · Authors · 2023-08-20
> > >
> > > Dear Reviewer J7TL,
> > >
> > > We greatly appreciate you for your additional comments. We would like to further explain the concerns below.
> > >
> > > **Concern #1: The analysis is done on a toy dataset that exhibits only one spurious feature.**
> > >
> > > **Response:** We conduct experiments on MNIST-LT, Fashion-LT, CIFAR-LT, and ImageNet-LT, and a manually designed benchmark CMNIST-LT. As we also replied to Reviewer qgru, we design CMNIST-LT to reveal the insights from a simple case. Moreover, **the other four are real-world datasets**, and we conduct multiple empirical studies on these four datasets in Table 1 and Figure 3 in the main paper. The results demonstrate the negative effects of spurious features on long-tailed datasets.
> > >
> > > **Concern #2: The proposed method lacks novelty wrt [28].**
> > >
> > > **Response:** Sorry for the incomplete previous response. We would like to explain that our method is different from FSA[28]. **FSA is a two-stage method**, which applies the augmentation only to finetune the model in phase 2 (The training epochs of phase 2 are much less than phase 1). In phase 1, FSA learns the feature representation in a conventional manner. However, we find that re-sampling can benefit long-tail representation learning in the **single-stage** framework if the spurious correlation is avoided, and conduct empirical studies to demonstrate our findings. Moreover, our augmentation method is also different from FSA, and our method outperforms FSA by a large margin, as is discussed in the previous response.
> > >
> > > **Concern #3: The method does not achieve SOTA numbers.**
> > >
> > > **Response:** As is discussed in Section "Broader Impact and Limitations" in the main paper, our proposed module can not achieve comparable performance with the well-designed models, since our intention is not to achieve performance that is on par with state-of-the-art methods. Instead, we hope that our findings will inspire future research regarding re-sampling methods. Moreover, our method can achieve high performance by combining SOTA losses such as BCL loss, which is reported in the previous response.

---

> > > > ### Comment · Reviewer_J7TL · 2023-08-21
> > > >
> > > > Dear authors,
> > > >
> > > > thank you for the additional comments.
> > > > My concerns are addressed, I've updated my rating.
> > > >
> > > > However, the authors should update tables with the most recent methods in the field and combine their method with BCL (at least) for all the dataset to achieve close to sota numbers. As the current implementation lags far behind especially for ImageNet-LT.

---

> > > > > ### Author Response · Authors · 2023-08-21
> > > > >
> > > > > Dear Reviewer J7TL,
> > > > >
> > > > > Thank you a lot for your time and effort in the reviews, and following your kind suggestions, we will update the results in the next version.
> > > > >
> > > > > Best Regards,
> > > > >
> > > > > Authors

---

> ### Author Response · Authors · 2023-08-18
>
> Dear Reviewer J7TL:
>
> Thank you again for your time and thoughtful comments! We hope our additional results and responses could address your concerns. Since the discussion period is about to close, we would appreciate it if you would kindly let us know of any further questions. We would be always available to clarify any of your questions.
>
> Best regards,
>
> Yours sincerely authors

---

### Official Review · Reviewer_qgru · 2023-07-02

**Soundness:** 3 good
**Presentation:** 2 fair
**Contribution:** 3 good
**Rating:** 5
**Confidence:** 4

**Summary:**

In this paper, the authors rethink the effect of re-sampling on long-tailed learning. Specifically, this paper leverages a synthetic experiment to claim that re-sampling is sensitive to irrelevant contexts.
To address this problem, this paper proposes a context-shift augmentation strategy to alleviate the overfitting to the irrelevant contexts in the tail classes. In the empirical part, this paper conducts performance comparisons on CIFAR-LT and ImageNet-LT to validate the proposed method.


**Strengths:**

- The paper is generally well-written and easy to follow.
- The exploration on the intrinsic mechanism of re-sampling on long-tailed data is important and not addressed.
- This paper claims that re-sampling is sensitive to the irrelevant context and presents a toy example to support the claim.


**Weaknesses:**

- The construction of CMNIST-LT seems an extreme case, which may not exhibit in real-world scenarios.
- The proposed method still follows the multi-branch architecture. However, the computational cost of the proposed method is not discussed.
- The experiments are not adequate. Some important long-tailed benchmarks or baselines are missing.
- This paper lacks more ablation studies or further analysis to provide more insight of the proposed baseline.


**Questions:**

- The construction of CMNIST-LT seems unreasonable. The color injection introduces severe spurious correlations to the tail classes as each tail class is assigned with single color and each head class is assigned with multiple colors. It is not surprising that the class-balanced re-sampling will deteriorate the performance at this case as only the spurious feature (single color) is enough to perform the classification on the tail class. Furthermore, this hand-crafted distribution on the irrelevant contexts may not exhibit on CIFAR-10/100-LT, ImageNet-LT as the diversity of the tail-class irrelevant contexts is mainly proportional to their class cardinality, but rather than collapse to the single point in the distribution.
- The author raises the question “Can re-sampling benefit long-tail learning in the single-stage framework” in line 41. However, the proposed method still requires multi-branch learning (even multi-stage learning) to perform the resampling. It is critical to compare the computational cost of the proposed method with baseline methods.
- This paper mainly investigates the class-balanced re-sampling. Is this re-sampling strategy optimal at the case of irrelevant contexts? More discussions or empirical comparisons are recommended.
- Intuitively, the Grad-CAM is not reliable at the early stages of the training. It seems multiple stages (warmup) are required to ensure the training stability.
- The experiments are only conducted on CIFAR-LT and ImageNet-LT. It would be better to conduct experiments on more real-world long-tailed benchmark Places-LT and iNaturalist.
- In the main text of this paper, only the empirical comparison results are presented. It would benefit a lot to provide more ablation studies or further analysis of the proposed method.
- Some supervised long-tailed learning SOTA baselines are missing in this paper:

[1] Menon, A. K., Jayasumana, S., Jain, H., Veit, A., Kumar, S., & Rawat, A. S. Long-tail learning via logit adjustment. ICLR 2020.

[2] Cui, J., Zhong, Z., Liu, S., Yu, B., & Jia, J. (2021). Parametric contrastive learning. ICCV 2021.

- Some related data augmentation baselines are missing in this paper:

[3] Chu, P., Bian, X., Liu, S., & Ling, H. (2020). Feature space augmentation for long-tailed data. ECCV 2020.

[4] Li, S., Gong, K., Liu, C. H., Wang, Y., Qiao, F., & Cheng, X. (2021). Metasaug: Meta semantic augmentation for long-tailed visual recognition. CVPR 2021.

[5] Zang, Y., Huang, C., & Loy, C. C. (2021). Fasa: Feature augmentation and sampling adaptation for long-tailed instance segmentation. ICCV 2021.

[6] Zhou, Z., Yao, J., Wang, Y. F., Han, B., & Zhang, Y. (2022). Contrastive learning with boosted memorization. ICML 2022.

[7] Ahn, S., Ko, J., & Yun, S. Y. (2023). CUDA: Curriculum of Data Augmentation for Long-tailed Recognition. ICLR 2023.


**Limitations:**

Yes.

---

> ### Author Rebuttal · Authors · 2023-08-09
>
> We are grateful for the thoughtful reviews, and for the comments that the paper is 'well-written'. We will address the concerns below.
>
> **Concern #1 (Weakness #1 & Question #1): The construction of CMNIST-LT seems an extreme case and unreasonable.**
>
> **Response:** Thanks for your concern. The CMNIST-LT benchmark is inspired by the widely-used CMNIST [1-4], which is also designed to simulate biased data. Your understanding is correct since the injected color is the single factor that affects the learning results. In real-world data, the class-balanced re-sampling (CB-RS) will also deteriorate the performance if some spurious features are enough to perform the classification on the tail class. It might be not surprising, but we just intend to reveal the insights from a simple case. Moreover, we also validate this idea by conducting empirical studies on real-world datasets CIFAR-LT and IMAGENET-LT in Table 1 and Figure 3 in the main paper.
>
> **Concern #2 (Weakness #2 & Question #2): The computational cost of the proposed method is not discussed.**
>
> **Response:** Our method is based on the widely-used dual-branch network. Moreover, we add a context-extracting module to calculate Grad-CAM and get the backgrounds. This module has a very small computational cost, as it only needs to apply the gradient backward once at the last layer of the network. Compared to the global gradient backward for updating the whole model, this cost can be ignored. Also, we do not need too much space to save the extracted backgrounds, since we set the size of the context bank equal to the mini-batch size. Moreover, we calculate the time cost per epoch of CIFAR100-LT using a single RTX3090. Following are the results.
>
> | Loss          | CE     | BCL    |
> | ------------- | ------ | ------ |
> | Single-Branch | 2.04 s | 4.76 s |
> | Dual-Branch   | 2.38 s | 4.85 s |
> | Ours          | 2.98 s | 5.10 s |
>
> **Concern #3 (Weakness #3 & Questions #5&7): More long-tailed benchmarks or baselines.**
>
> **Response:** Thanks for your suggestion. We have combined our method with LA and reported the results in Section C.7 in the supplementary material. Following are the experimental results.
>
> |            | Overall | Many | Medium | Few  |
> | ---------- | ------- | ---- | ------ | ---- |
> | Ours       | 45.8    | 64.3 | 49.7   | 18.2 |
> | Ours w/ LA | 47.2    | 62.4 | 48.8   | 26.1 |
>
> Moreover, our proposed Context-Shift Augmentation (abbreviated as CSA) can be integrated with the contrastive-learning-based method BCL to further improve the generalization. Following are the experimental results.
>
> | CIFAR100-LT         | ir=100 | ir=50 | ir=10 |
> | ------------------- | ------ | ----- | ----- |
> | BCL                 | 51.9   | 56.6  | 64.9  |
> | BCL w/ proposed CSA | 52.6   | 57.1  | 65.8  |
>
> We conduct experiments on Places-LT and report the results below. The results show that our method outperforms cRT.
>
> |      | Overall | Many | Medium | Few  |
> | ---- | ------- | ---- | ------ | ---- |
> | cRT  | 36.7    | 42.0 | 37.6   | 24.9 |
> | Ours | 37.2    | 42.3 | 38.7   | 22.4 |
>
> **Concern #4 (Weakness #4 & Question #6): More ablation studies or further analysis.**
>
> **Response:** We have done ablation studies and further analysis. The results are reported in the supplementary material due to the page limit. Please refer to Section C for more details.
>
> **Concern #5 (Question #3): Is this re-sampling strategy optimal at the case of irrelevant contexts?**
>
> **Response:** There might be some misunderstanding. It is not that using re-sampling can solve irrelevant contexts, but that directly using re-sampling can cause overfitting to the irrelevant contexts. So we propose a context-shift augmentation method to alleviate this problem.
>
> **Concern #6 (Question #4): The Grad-CAM is not reliable at the early stages of the training. Multiple stages (warmup) are required to ensure the training stability.**
>
> **Response:** We have multiple strategies to ensure training stability. First, we have a warmup stage of 10 epochs for all experiments. The implementation details can be referred from Section B in the supplementary material. Second, we use a threshold $\delta$ to filter only the well-learned training samples to ensure the quality of extracted backgrounds, as is mentioned in Section 3.1. The validation experiment of $\delta$ can be found in Section C.1 in the supplementary material.
>
> **Concern #7 (Question #8): Some related data augmentation baselines are missing in this paper.**
>
> **Response:** We have already compared and discussed some data augmentation baselines such as FSA [5] in Sections 3.3&4.3. The accuracy on IMAGENET-LT is 42.7 (ours) vs. 35.2 (FSA). Besides, we will add the other given literature [6-9] in the next version. Thanks for your notation.
>
> [1] Invariant Risk Minimization.
>
> [2] Learning Not to Learn: Training Deep Neural Networks with Biased Data.
>
> [3] REPAIR: Removing Representation Bias by Dataset Resampling.
>
> [4] Learning from Failure: Training Debiased Classifier from Biased Classifier.
>
> [5] Chu, P., Bian, X., Liu, S., & Ling, H. (2020). Feature space augmentation for long-tailed data. ECCV 2020.
>
> [6] Li, S., Gong, K., Liu, C. H., Wang, Y., Qiao, F., & Cheng, X. (2021). Metasaug: Meta semantic augmentation for long-tailed visual recognition. CVPR 2021.
>
> [7] Zang, Y., Huang, C., & Loy, C. C. (2021). Fasa: Feature augmentation and sampling adaptation for long-tailed instance segmentation. ICCV 2021.
>
> [8] Zhou, Z., Yao, J., Wang, Y. F., Han, B., & Zhang, Y. (2022). Contrastive learning with boosted memorization. ICML 2022.
>
> [9] Ahn, S., Ko, J., & Yun, S. Y. (2023). CUDA: Curriculum of Data Augmentation for Long-tailed Recognition. ICLR 2023.

---

> > ### Comment · Reviewer_qgru · 2023-08-16
> > **Thanks for the rebuttal**
> >
> > Dear authors,
> >
> > Thanks for the detailed replies to my concerns. After reading the rebuttal and other reviews, my concerns have been adequately addressed. I would like to raise the score to '5'.
> >
> > Best,
> >
> > Reviewer qgru

---

### Official Review · Reviewer_gDP8 · 2023-07-05

**Soundness:** 3 good
**Presentation:** 3 good
**Contribution:** 3 good
**Rating:** 7
**Confidence:** 4

**Summary:**

This article systematically investigates the effectiveness of resampling methods in long-tailed image classification problems. Resampling is a classical and important approach to addressing class imbalance, but it may not always be effective when applied to long-tailed datasets. The paper shed light on this phenomenon and provides a systematical analysis. It discovers that re-sampling the tail data can improve the performance, but may lead to overfitting on the irrelevant contexts. To address this issue, they propose a context-shift data augmentation method. The proposed method achieves superior performance on long-tailed datasets such as CIFAR10-LT, CIFAR100-LT, and ImageNet-LT.

**Strengths:**

1. This paper provides a well-defined and reasonable motivation. The authors show both success and failure cases of re-sampling on long-tailed datasets, which highlights the significance of addressing this problem.
2. This paper extensively conducts experiments and ablation studies on widely-used datasets, which demonstrates the effectiveness of their method.
3. This paper is well-writing and easy to follow. The source code is provided in a proper way, facilitating the reproducibility of their work.


**Weaknesses:**

1. A comparison with some data augmentation methods should be included. For example, Randaugment is a widely used method, which can improve the performance on long-tailed datasets.
2. The name of the proposed method (context-shift augmentation) should be unified, as some parts include hyphens "-" while others do not. Also, some parts use italics while others do not.


**Questions:**

1. Does the data augmentation method of enhancing the background also applied to samples of the head class?
2. Does the proposed context-shift augmentation require additional computational cost?


**Limitations:**

The authors have addressed the limitations.

---

> ### Author Rebuttal · Authors · 2023-08-09
>
> We appreciate the reviewer for the comments and for mentioning that this paper has a 'well-defined' and 'reasonable' motivation. We address the concerns below.
>
> **Concern #1 (Weakness #1): Comparison with some data augmentation methods such as Randaugment.**
>
> **Response:** Thanks for your suggestion. AutoAugment [1] and Cutout [2] can be applied to further improve the performance. Following are the results.
>
> | CIFAR100-LT (ir=100) | Overall | Many | Medium | Few  |
> | -------------------- | ------- | ---- | ------ | ---- |
> | CE                   | 44.1    | 72.5 | 45.1   | 7.7  |
> | cRT                  | 49.4    | 66.7 | 50.7   | 26.3 |
> | Ours                 | 50.4    | 64.6 | 55.7   | 26.5 |
>
> Moreover, Randaugment is usually used in contrastive learning methods. Our proposed Context-Shift Augmentation (abbreviated as CSA) can be integrated with the contrastive-learning-based method BCL to further improve the generalization. The results are also shown as follows.
>
> | CIFAR100-LT         | ir=100 | ir=50 | ir=10 |
> | ------------------- | ------ | ----- | ----- |
> | BCL                 | 51.9   | 56.6  | 64.9  |
> | BCL w/ proposed CSA | 52.6   | 57.1  | 65.8  |
>
> **Concern #2 (Weakness #2): The name of the proposed method should be unified.**
>
> **Response:** Thanks for your kind suggestion. We will revise the miswriting in the next version.
>
> **Concern #3 (Question #1): Does the data augmentation method of enhancing the background also applied to samples of the head class?**
>
> **Response:** Yes, we extract backgrounds from the uniform sampling and apply it to the balanced sampling. In this manner, all samples are used for augmentation.
>
> **Concern #4 (Question #2): Does the proposed context-shift augmentation require additional computational cost?**
>
> **Response:** Our method is based on the widely-used dual-branch network. Apart from this, we add a context-extracting module to calculate Grad-CAM and get the backgrounds. This module has a very small computational cost, as it only needs to apply the gradient backward once at the last layer of the network. Compared to the global gradient backward for updating the whole model, this cost can be ignored. Also, we do not need too much space to save the extracted backgrounds, since we set the size of the context bank equal to the mini-batch size. Moreover, we calculate the time cost per epoch of CIFAR100-LT using a single RTX3090. Following are the results.
>
> | Loss          | CE     | BCL    |
> | ------------- | ------ | ------ |
> | Single-Branch | 2.04 s | 4.76 s |
> | Dual-Branch   | 2.38 s | 4.85 s |
> | Ours          | 2.98 s | 5.10 s |
>
> [1] Autoaugment: Learning augmentation strategies from data.
>
> [2] Improved regularization of convolutional neural networks with cutout.

---

> > ### Comment · Reviewer_gDP8 · 2023-08-20
> >
> > Thanks for the author's response. I would also maintain my initial score of accept.

---

### Official Review · Reviewer_gT6X · 2023-07-17

**Soundness:** 3 good
**Presentation:** 3 good
**Contribution:** 3 good
**Rating:** 6
**Confidence:** 4

**Summary:**

Long-tailed Learning has drawn great attention in recent development of deep learning due to the non-negligible challenges on tail classes during learning. The re-sampling method that should make an effect to improve the training usually fails in a lot of practices. The authors dive into the phenomenon and finds some new insights towards this problem, and then design a new method to alleviate the negative impact of re-sampling on some datasets, which shows the final promise compared with the recent SOTAs.

**Strengths:**

The strengths of this submission can be summarized as follows,

1) The authors rethinks an interesting problem about the re-sampling method, which are under-estimated or mistaken in previous studies. That is, re-sampling mainly works in the classifier layer through a two-state decoupling way. How to make the re-sampling works in a single-stage framework is an unexplored yet important challenge.

2) The observation and interpretation are insightful, as two different types of datasets contracts on the final performance for re-sampling. It means actually re-sampling can help to extract the discriminative representation but just sensitive to the irrelevant contexts, i.e., overfitting to the spurious correlations in some cases.

3) Based on the conjecture, the authors design a context-shift augmentation method along with re-sampling to alleviate the negative impact. Specially, they construct a context memory bank to boost the augmentation of samples in diversity and avoid overfitting the spurious correlation from backgrounds. The experimental results demonstrate the promise of the proposed method.

4) The whole paper is well written, in which the motivation parts are well justified with a range of experiments to show the intuition, making the whole parts clearly understood.

**Weaknesses:**


Although the submission are well organized and presented, there are still some points that can be improved by following the below advices.

1) In Figure 2, although it is hard to visual the representation distribution on CIFAR100-LT and ImageNet-LT, it is still possible to randomly select some sub-classes, e.g., downsample 10 classes from the super classes of CIFAR100-LT, to visualize the representation. This can shows whether on some datasets that contains many class-irrelevant contexts, the representation are not well learnt, i.e., not so discriminative, compared with that on datasets that contains few class-irrelevant contexts.

2) Similarly, on basis of Figure 3, it is helpful to show with the context-shift augmentation, the method can efficiently avoid overfitting the background spurious correlations, by tracing the Grad-CAM map of some same examples.

3) It will be more solid to give some interpretation from the theoretical analysis: why the re-sampling encourages the method to learn more spurious correlations compared the method without re-sampling, or why the two-stage methods reduce the overfitting of spurious correlation in re-sampling. What the benefits or lessons we get in re-sampling from some theoretical perspectives.

4) It lacks of more experiments about some hyperparameters like \lambda of Eq.(7). Especially, the hyperparameter can be tuned based on different datasets and even different classes. The authors can strengthen this parts to avoid the concerns about the hyperparameter selection or heuristics.

**Questions:**


1) How is the visualization of representation on datasets that contains more class-irrelevant contexts with or without resampling? And how is the visualization of Grad-CAM for examples with or without context-shift augmentation?

2) How is the sensitivity of the hyparameter lambda in Eq. (7)? How is the performance of the proposed method applied to some recent SOTAs, i.e., their compatibility with recent SOTAs?

 3) Why do not consider the SOTAs like LA or BCL, PaCo in the experimental parts? The authors can have some more explanations in choosing the baselines.

---

> ### Author Rebuttal · Authors · 2023-08-09
>
> We appreciate the reviewer for the thoughtful reviews, and for noting that the observation and interpretation are 'insightful', and the whole paper is 'well written'. We will address the concerns below.
>
> **Concern #1 (Weakness #1): Visualize the representation distribution on CIFAR100-LT.**
>
> **Response:** Following your suggestion, we visualize the representation of CIFAR100-LT in Figure (r1) in the Rebuttal PDF file. Each color represents a class, with darker colors representing head classes, and lighter colors representing tail classes. Although the fine-grained colors make it difficult to distinguish some classes, it can still be seen that CB-RS learns worse representation compared with vanilla CE. Moreover, our proposed method can learn a more distinguishable representation.
>
> **Concern #2 (Weakness #2 & Question #1): Visualize the Grad-CAM for examples with context-shift augmentation.**
>
> **Response:** We follow your suggestion and visualized the Grad-CAM for examples with our proposed context-shift augmentation. We choose the same examples in Figure 3 in the main paper, and report the results in Figure (r2) in the Rebuttal PDF file. The results show that our method can alleviate the negative impact on head-class samples caused by the overfitting problem.
>
> **Concern #3 (Weakness #3): Give some interpretation from the theoretical analysis.**
>
> **Response:** Thank you for your suggestions. We conduct our research mainly from an empirical perspective since our work is the first to study the reason for overfitting caused by re-sampling. It is a good suggestion to analyze from the theoretical aspect, and we will leave this for future research.
>
> **Concern #4 (Weakness #4  & Question #2): Experiments about some hyper-parameters like $\lambda$ of Eq.(7).**
>
> **Response:** We have done experiments about hyperparameters. We report the results in the supplementary material due to the page limit. Please refer to Section C for more details.
>
> **Concern #5 (Question #2 & Question #3): Consider more recent SOTAs.**
>
> **Response:** We have combined our method with LA and reported the results in Section C.7 in the supplementary material. Following are the experimental results.
>
> |            | Overall | Many | Medium | Few  |
> | ---------- | ------- | ---- | ------ | ---- |
> | Ours       | 45.8    | 64.3 | 49.7   | 18.2 |
> | Ours w/ LA | 47.2    | 62.4 | 48.8   | 26.1 |
>
> Moreover, our proposed Context-Shift Augmentation (abbreviated as CSA) can be integrated with the contrastive-learning-based method BCL to further improve the generalization. Following are the experimental results.
>
> | CIFAR100-LT         | ir=100 | ir=50 | ir=10 |
> | ------------------- | ------ | ----- | ----- |
> | BCL                 | 51.9   | 56.6  | 64.9  |
> | BCL w/ proposed CSA | 52.6   | 57.1  | 65.8  |

---

> > ### Comment · Reviewer_gT6X · 2023-08-16
> >
> >
> > Thank you for the authors' detailed response. After reading the rebuttal and the comments of the other reviewers, my concerns have been thoroughly addressed and thus I maintain towards the acceptance of this work.
> >
> > Reviewer gT6X

---

### Official Review · Reviewer_jD4D · 2023-07-24

**Soundness:** 4 excellent
**Presentation:** 3 good
**Contribution:** 3 good
**Rating:** 5
**Confidence:** 4

**Summary:**

This paper studies the effect of re-sampling for long-tail learning. It finds that re-sampling is highly related with the contexts of images. And it does not necessarily work of fail in long-tail learning. Based on these findings, a context-shift augmentation module is proposed. Experiments verify the effectiveness of the proposed module.

**Strengths:**

The paper studies an important task and proposes a unique solution for the found issues of re-sampling.

The paper is well-written and easy to follow.

The paper comprehensively introduces the preliminary of the proposed method.

**Weaknesses:**

The concept of context and content appear a lot in the paper, but there is no formal introduction or definition for these two concepts.

The paper illustrates that the irrelevant context issue occurs for MNIST and Fashion. And for IMAGENET, the issue is not significant. So is the proposed method more suitable for those simple datasets like MNIST and Fashion? If yes, this will highly affect the application of the proposed method for real-world tasks.

The paper uses Grad-CAM to determine the foreground and background for training images. However, for tail-class images, since the model may baise to head class, the generated activation map may not be accurate, which will affect the final results.

**Questions:**

More examples are encouraged to be attached in Introduction to better illustrate the found issue and the basic motivation of the paper.

The paper illustrates that the irrelevant context issue occurs for MNIST and Fashion. And for IMAGENET, the issue is not significant. So is the proposed method more suitable for those simple datasets like MNIST and Fashion? If yes, this will highly affect the application of the proposed method for real-world tasks.

**Limitations:**

Yes

---

> ### Author Rebuttal · Authors · 2023-08-09
>
> We thank the reviewer for the comments, and are encouraged the reviewer comments that the paper is 'well-written'. We will address the concerns below.
>
> **Concern #1 (Weakness #1 & Question #1): There is no formal introduction or definition for context and content.**
>
> **Response:** Context refers to the semantically unrelated parts in the images, and content refers to the semantically related parts. We give an example in Figure (r4) in the Rebuttal PDF file. Thank you for pointing out this concern. We will revise the miswriting and standardize the definitions in the next version.
>
> **Concern #2 (Weakness #2 & Question #2): The irrelevant context issue occurs only for simple datasets like MNIST and Fashion. And for IMAGENET, the issue is not significant.**
>
> **Response:** There might be some misunderstanding about this concern. The irrelevant context issue **does not occur** for simple datasets like MNIST-LT and Fashion-LT, because training samples and target labels on these datasets are highly semantically correlated (even do not contain any backgrounds). Meanwhile, the irrelevant context issue **occurs** for real datasets like CIFAR-LT and IMAGENET-LT, because these datasets contain many semantically unrelated elements, such as backgrounds and irrelevant objects. In Table 1, both cRT (classifier re-training) and CB-RS (class-balanced re-sampling) are classical learning methods for long-tail learning. CB-RS learns well on MNIST-LT and Fashion-LT but fails on CIFAR-LT and IMAGENET-LT, demonstrating that the irrelevant context has negative impacts. So we propose a context-shift augmentation to alleviate the problem. Experimental results on CIFAR-LT and IMAGENET-LT validate the effectiveness of our proposed method.
>
> **Concern #3 (Weakness #3): The generated activation map of tail-class images may not be accurate.**
>
> **Response:** First, we generate the activation maps from the uniform sampling of the long-tailed data. In this sampling manner, most data are from head classes, and tail-class images are rarely sampled. Second, we use a threshold $\delta$ to filter the well-learned training samples to ensure the quality of extracted backgrounds, as is mentioned in Section 3.1. Moreover, we visualize some tail-class samples with predicted probabilities higher than $\delta$ in Figure (r3) in the Rebuttal PDF file, which shows that the model can still grap accurate activation maps for tail-class samples.

---

> ### Author Response · Authors · 2023-08-18
>
> Dear Reviewer jD4D:
>
> Thank you again for your time and thoughtful comments! We hope our additional results and responses could address your concerns. Since the discussion period is about to close, we would appreciate it if you would kindly let us know of any further questions. We would be always available to clarify any of your questions.
>
> Best regards,
>
> Yours sincerely authors

---

### Author Rebuttal · Authors · 2023-08-09

Dear Reviewers,

We greatly appreciate all of you for your thoughtful comments and valuable suggestions. These are very helpful for improving our paper. We have carefully referred to the concerns and written the response. In addition to the text responses, **we also report some figure results in the PDF file**. We hope the responses would meet your requirements.

Best Regards,

Authors

---

### Author Response · Authors · 2023-08-14
**We look forward to receiving your further feedback**

Dear Reviewers,

We would like to appreciate you again for your valuable comments. Moreover, it is important for us to know whether our responses have addressed your concerns, and we look forward to receiving your further feedback. In any case, we remain available to answer your questions.

Best Regards,

Authors

---

### Comment · Area_Chair_ah6h · 2023-08-18
**Reviewer-Author Discussion Period**

Dear All,

Thank you reviewers for your hard work in evaluating this submission, and thank you authors for responding to the reviewers’ questions and concerns.

We are now entering the final phase of the discussion period, which will run until 21 Aug, and some of the authors' responses have to been acknowledged by all reviewers.

@Reviewers, if you have any follow up questions or comments on the rebuttal or the responses, now is the time to express them. At the very least, please acknowledge that you have read the authors’ response to your review.

Thank you everyone for making the review process a fruitful, constructive, and civil process.

AC

---

### Decision · Program_Chairs · 2023-09-21

**Decision:**

Accept (poster)

**Comment:**

This paper focuses on long-tailed learning, which has drawn great attention in recent development of deep learning due to the non-negligible challenges on tail classes during learning. The re-sampling method that should make an effect to improve the training usually fails in a lot of practices. The authors dive into the phenomenon and finds some new insights towards this problem, and then design a new method to alleviate the negative impact of re-sampling on some datasets, which shows the final promise compared with the recent SOTAs.

Specifically, the authors rethinks an interesting problem about the re-sampling method, which are under-estimated or mistaken in previous studies. That is, re-sampling mainly works in the classifier layer through a two-state decoupling way. How to make the re-sampling works in a single-stage framework is an unexplored yet important challenge. The observation and interpretation are insightful, as two different types of datasets contracts on the final performance for re-sampling. It means actually re-sampling can help to extract the discriminative representation but just sensitive to the irrelevant contexts. The authors design a context-shift augmentation method along with re-sampling to alleviate the negative impact. They construct a context memory bank to boost the augmentation of samples in diversity and avoid overfitting the spurious correlation from backgrounds. The experimental results demonstrate the promise of the proposed method. The whole paper is well written, in which the motivation parts are well justified with a range of experiments to show the intuition, making the whole parts clearly understood.

Overall, the clarity and novelty are clearly above the bar of NeurIPS. While the reviewers had some concerns, the authors did a particularly good job in their rebuttal. Thus, all of us have agreed to accept this paper for publication! Please include the additional experimental results in the next version.